# Humoral Immunity of Unvaccinated COVID-19 Recovered vs. Naïve BNT162b2 Vaccinated Individuals: A Prospective Longitudinal Study

**DOI:** 10.3390/microorganisms11071628

**Published:** 2023-06-22

**Authors:** Gili Joseph, Carmit Cohen, Carmit Rubin, Havi Murad, Victoria Indenbaum, Keren Asraf, Yael Weiss-Ottolenghi, Gabriella Segal-Lieberman, Yitshak Kreiss, Yaniv Lustig, Gili Regev-Yochay

**Affiliations:** 1The Sheba Pandemic Preparedness Research Institute (SPRI), and Infection Prevention & Control Unit, Sheba Medical Center, Tel Hashomer, Ramat Gan 52621, Israel; gili.joseph@sheba.health.gov.il (G.J.); cohencvet@gmail.com (C.C.); yael.ottolenghi@sheba.health.gov.il (Y.W.-O.); 2Gertner Institute, Sheba Medical Center, Tel Hashomer, Ramat Gan 52621, Israel; carmit.rubin@sheba.health.gov.il (C.R.); havim@gertner.health.gov.il (H.M.); 3Central Virology Laboratory, Public Health Services, Ministry of Health, Tel-Hashomer, Ramat Gan 52621, Israel; viki.indenbaum@sheba.health.gov.il (V.I.); yaniv.lustig@sheba.health.gov.il (Y.L.); 4The Dworman Automated-Mega Laboratory, Sheba Medical Center, Tel-Hashomer, Ramat Gan 52621, Israel; keren.asraf@sheba.health.gov.il; 5Division of Endocrinology, Diabetes and Metabolism, Sheba Medical Center, Tel Hashomer, Ramat Gan 52621, Israel; gabriella.lieberman@sheba.health.gov.il; 6Sackler School of Medicine, Tel-Aviv University, Tel Aviv 69978, Israel; 7General Management, Sheba Medical Center, Tel Hashomer, Ramat Gan 52621, Israel; yitshak.kreiss@sheba.health.gov.il

**Keywords:** immunoglobulin G, antibodies, neutralizing, immunity, humoral, COVID-19, infection, BNT162b2 vaccine

## Abstract

To study the differences in the immune response to SARS-CoV-2 infection compared to the response to vaccination, we characterized the humoral immune kinetics of these situations. In this prospective longitudinal study, we followed unvaccinated COVID-19-recovered individuals (*n* = 130) and naïve, two-dose BNT162b2-vaccinated individuals (*n* = 372) who were age- and BMI-matched for six months during the first pandemic year. Anti-RBD-IgG, neutralizing antibodies (NAbs), and avidity were assessed monthly. For recovered patients, data on symptoms and the severity of the disease were collected. Anti-RBD-IgG and NAbs titers at peak were higher after vaccination vs. after infection, but the decline was steeper (peak log IgG: 3.08 vs. 1.81, peak log NAbs: 5.93 vs. 5.04, slopes: −0.54 vs. −0.26). Peak anti-RBD-IgG and NAbs were higher in recovered individuals with BMI > 30 and in older individuals compared to individuals with BMI < 30, younger population. Of the recovered, 42 (36%) experienced long-COVID symptoms. Avidity was initially higher in vaccinated individuals compared with recovered individuals, though with time, it increased in recovered individuals but not among vaccinated individuals. Here, we show that while the initial antibody titers, neutralization, and avidity are lower in SARS-CoV-2-recovered individuals, they persist for a longer duration. These results suggest differential protection against COVID-19 in recovered-unvaccinated vs. naïve-vaccinated individuals.

## 1. Introduction

The coronavirus disease 2019 (COVID-19) pandemic has caused a global health crisis with great morbidity and mortality. By March 2023, almost 7 million people had died from SARS-CoV-2, and more than 750 million had recovered [1]. mRNA vaccines were developed by BioNTech, Pfizer, and Moderna and distributed starting in December 2020 to stop the spread of the disease [2]. However, despite their effectiveness [3,4,5], a waning of the antibody titer was observed after about six months, increasing the risk of breakthrough infections [6,7,8,9,10,11]. Studies investigating correlates of protection have found that SARS-CoV-2 receptor binding domain (RBD), immunoglobulin G (IgG), and specifically neutralizing antibodies are correlated with protection from breakthrough infections [4,12]. IgG titers remain high for approximately four months following a second Pfizer vaccine dose [6,13,14,15], with differences according to patient characteristics, such as age, sex, and BMI [16,17,18,19,20]. Examining the humoral immune response to SARS-CoV-2 infection in recovered COVID-19 patients was mostly confined to the early stages of recovery and rarely followed the immune response dynamics [16,17,18,19,20]. Up until the emergence of the Omicron VOC, rates of reinfection by SARS-CoV-2 recovered patients were significantly lower than vaccine breakthrough cases among vaccinated individuals [21,22,23]. This suggests a different immune response following vaccination and infection.

Thus, comparing the immune response dynamics induced by infection to those induced by vaccination is highly important. At present, most of the world’s population has acquired hybrid immunity, i.e., a combination of vaccination and infection, so studying these differences is currently impossible. Here we report a six-month prospective, longitudinal study that was initiated in the early days of the pandemic, comparing unvaccinated individuals, infected with the ancestral SARS-CoV2, to naïve individuals who received two doses of the BNT162b2 vaccine.

## 2. Materials and Methods

### 2.1. Study Design and Population

The study was conducted from 25 March 2020 to 26 April 202, when the first patients were admitted to Sheba Medical Center (SMC). Patients diagnosed with SARS-CoV-2 as detected by reverse transcriptase-polymerase chain reaction (RT-PCR) who agreed to participate and give a blood sample on a monthly basis and signed an informed consent form were recruited. The recovered volunteer patients were followed monthly for one year after SARS-CoV-2 detection. Follow-up included serology tests for SARS-CoV-2 anti-RBD IgG (anti-RBD-IgG) and neutralizing antibodies (NAbs). The recovered volunteers were also asked to answer epidemiological and medical questionnaires presented at the time of recruitment and a medical follow-up questionnaire on every visit (Appendix A). From the 180 enrolled patients, 130 had more than three longitudinal samples and completed at least six months of survey. From this cohort, we analyzed the avidity index for 17 individuals that had complete data at one month, six months, and a year after recovery (Figure 1).

In order to evaluate differences in humoral response kinetics between recovered patients and BNT162b2-vaccinated health care workers (HCW), we matched 1–4 individuals from the SMC HCW-vaccinated cohort to each of the recovered patients with data encompassing six months’ survey or over. Cohorts were matched by the availability of the NAbs assay, the number of records, age, BMI, and the availability of results one month and six months post-vaccination (we did not match cases by sex owing to the skewness of SMC HCW towards females). From the 130 recovered individuals, only 120 could be matched to BNT162b2-vaccinated individuals.

We selected age- and BMI-matched vaccinated individuals, from a cohort of 4868 vaccinated HCW with two doses of BNT162b2. A 3:1 matching of vaccinated to recovered patients was conducted (range = 1–6 matched vaccinated per recovered), yielding a total cohort of 372 vaccinated individuals and 120 recovered patients. (Table 1). Data collected included sex, age, body mass index (BMI), comorbidities, and, among the recovered patients, infection severity and long-COVID symptoms (defined as persistent symptoms for >six weeks after recovery). Long-COVID symptoms were categorized as mental, neurologic, cardiovascular, respiratory, and others. Symptoms such as general fatigue, recurring muscle or joint pain, and nausea were jointly referred to as other symptoms.

The severity of infection was divided into asymptomatic, mild, moderate, and severe according to Clinical Management of COVID-19 WHO, Interim Guidance, May 2020 [24], (Appendix A): In general, mild disease was defined if a symptomatic patient met the case definition for COVID-19 without evidence of pneumonia or hypoxia. Moderate disease was defined if the individual had clinical signs of pneumonia (fever, cough, dyspnea, and lung infiltrates) but without significant hypoxia (SpO2 ≥ 90% on room air). Severe disease was defined if the patient had clinical signs of pneumonia (fever, cough, dyspnea, and fast breathing) plus one of the following: respiratory rate > 30 breaths/min; severe respiratory distress; or SpO2 < 90% on room air.

To characterize humoral response kinetics, we tested patients’ sera for RBD-binding IgG antibodies and the SARS-CoV-2 pseudo-virus neutralization assay on a monthly basis, and a subgroup was examined for avidity at 1, 6, and 12 months following recovery.

The protocol and informed consent were approved by the institution review board of the Sheba Medical Center. Written informed consent was obtained by all participants.

### 2.2. Serological Assays

Blood samples were centrifuged at 4000× *g* for 4 min at room temperature. Sera were tested for SARS-CoV-2 RBD IgG using the commercial automatic immunoassay access SARS-CoV-2 IgG (Beckman Coulter, CA, USA) according to the manufacturer’s instructions.

A SARS-CoV-2 pseudo-virus (psSARS-2) neutralization assay was performed as previously described [13] using a green fluorescent protein (GFP) reporter-based pseudotyped virus with a vesicular stomatitis virus (VSV) backbone coated with the SARS-CoV-2 spike (S) protein. Following titration, 100 focus-forming units of psSARS-2 were incubated with a twofold serial dilution of heat-inactivated (56 °C for 30 min) tested sera. After incubation for 60 min at 37 °C, the virus/serum mixture was transferred to Vero E6 cells (CRL-1586, ATCC) that had been grown to confluency in 96-well plates and incubated for 90 min at 37 °C. After the addition of 1% methyl cellulose (M0512, Sigma-Aldrich, St. Louis, MO, USA) in Dulbecco’s modified Eagle’s medium (Biological Industries) with 2% fetal bovine serum (Biological Industries, Kibbutz Beit-Haemek, Israel), plates were incubated for 24 h, and a 50% plaque reduction titer was calculated by counting green fluorescent foci using a fluorospot reader (AID Autoimmun Diagnostika, Straßberg, Germany). Sera not capable of reducing viral replication by 50% at 1:8 dilution or below were considered nonneutralizing. For clear presentation, nonneutralizing samples were marked with a titer of 2.

The avidity assay was based on an in-house RBD-IgG enzyme-linked immunosorbent assay with the addition of 6 M urea (U5378, Sigma-Aldrich) or PBS (Biological Industries) for 10 min for each sample [25]. Specifically, a 96-well microtiter Polysorb plate (Nunc, Thermo) was coated overnight at 4 °C with 50 μL per well of 1 μg ml−1 RBD antigen. After blocking with 5% skimmed milk at 25 °C for 60 min, serum samples diluted 1:100, 1:400, and 1:1000 with 3% skimmed milk were added to antigen-coated wells. The plate was incubated at 25 °C for 120 min, and following washing, each sample was incubated either with the addition of 6 M urea or PBS for 10 min. After washing, a goat anti-human IgG horseradish peroxidase conjugate (catalog 109–035–088, Jackson ImmunoResearch, West Grove, PA, USA) (diluted 1:15,000) was added to each well for 60 min. After washing, incubation of TMB Substrate Solution (Abcam, Cambridge, UK) for 5 min, and the addition of stop solution (2 N HCl), the optical density (OD) of each well was measured at 450 nm using a microplate reader (Sunrise, Tecan). The avidity index was calculated as the ratio (in percentage) between sample OD with 6 M urea and sample OD with PBS.

### 2.3. Statistical Analysis

Mixed-effects linear models were used to compare log-transformed IgG levels and neutralizing antibodies (NAbs) levels of matched vaccinated subjects and recovered subjects over time. The models included a random intercept for each matched set, and we allowed a first-order autoregressive (AR (1)) structure variance-covariance matrix for repeated measurements of the same subject. We fitted a model that included a vaccinated/recovered indicator, time in periods of 28 days (slope in recovered subjects), and interaction of time with the vaccinated/recovered indicator (difference of slopes in vaccinated and recovered subjects), adjusted for sex and comorbidity (a categorical variable including subjects with no comorbidities, one comorbidity, and more than two comorbidities). This will be called our basic model. Since the baseline measurement at time 0 was the peak IgG or peak NAbs, the regression coefficient of the vaccinated/recovered indicator is the difference in log-transformed peak values between vaccinated and recovered subjects. Graphs of population-based predicted values for vaccinated versus recovered subjects were generated from these models. The graphs show predicted values of log IgG for females with no comorbidities. The normality of the residuals from the models was verified.

To compare the IgG and NAb behavior of vaccinated and recovered subjects in different sub-groups (i.e., age sub-groups: subjects older versus younger than 45; BMI sub-groups: subjects with BMI > 30 vs. BMI < 30; and sex—males versus females), we added relevant interactions to the basic model. For example, to compare vaccinated/recovered behavior in older adults (>45) versus younger adults, we added age group and its second-order interactions with time and with the vaccinated/recovered indicator, as well as a third-order interaction between age, time, and the vaccinated/recovered indicator. We retained in the model only the significant interactions (*p* < 0.05). We also compared vaccinated/recovered behavior in BMI groups and in sex in a similar way.

We also examined whether the differences in peak levels of log values of IgG/NAbs between older versus younger than 45 subjects and between subjects with a BMI higher than 30 and lower than 30 were associated with different comorbidity levels. To do that, we used similar mixed models with additional interaction between age group/BMI group, and comorbidity level, separately in vaccinated and recovered subjects.

Additionally, a sub-analysis on recovered subjects only, with similar mixed linear models, was performed in order to examine whether the differences in peak levels of log values of IgG/NAbs between age groups and BMI groups were associated with different severity levels of COVID-19 symptoms (i.e., subjects were divided into groups of asymptomatic, mild, moderate, and severe symptoms). To do that, additional variables included in this sub-analysis were severity of symptoms, age, or BMI, respectively, and interaction terms between severity level and age/BMI groups. Another sub-analysis on recovered subjects only was conducted to examine the effects of long-COVID (a dichotomous variable of subjects who suffered long-COVID symptoms versus subjects without long-COVID symptoms) on the peak levels and on the slope with time by adding the main effect of long-COVID and its interaction with period to the model.

Geometric means (GMT) of avidity were calculated for recovered and vaccinated subjects and compared at different time points (1, 6, and 12 months after vaccination or recovery) using a two-sample student *t*-test.

## 3. Results

The study population included 492 individuals; 120 of them recovered from COVID-19 and were unvaccinated, and 372 were vaccinated twice with the BNT162b2 Pfizer vaccine (Table 1).

In order to compare the humoral immune response of recovered versus vaccinated individuals, the predicted values of log RBD—binding IgG and neutralizing Abs (NAbs)—were calculated from a mixed-linear model adjusted for sex and number of comorbidities, by period, for vaccinated versus recovered individuals (Appendix A). Figure 2A presents the predicted values by time period for vaccinated and recovered individuals. The plot shows that at the peak period (approximately 3 weeks after a positive COVID-19 result), log-transformed IgG values are higher for vaccinated individuals compared to recovered individuals (3.08 vs. 2.01; *p* < 0.0001). Antibody waning over time is observed in both recovered and vaccinated individuals, but waning is more rapid (a steeper decrease) in the vaccinated (slope of −0.54 vs. −0.26, respectively). Consequently, in the fourth period (113–140 days post-recovery/vaccination), the IgG values of recovered individuals are higher than those of vaccinated individuals. Similar behavior is presented for log-transformed NAbs (Figure 2B), although the differences between vaccinated and recovered are smaller (peak levels of 5.93 vs. 5.04 and slopes −0.29 vs. −0.17, respectively). Starting from the fifth period (141–168 days post-recovery/vaccination), the Nabs values of recovered individuals are higher than those of vaccinated individuals.

To evaluate the functionality of antibodies, we tested the RBD-binding IgG avidity score at one- and six-months post-vaccination and at 1, 6, and 12 months following recovery (Figure 2C). Avidity was calculated as the ratio (in percentage) between sample OD with 6 M urea and sample OD with PBS. One month after vaccination, avidity GMT was 43.53% (95% Cl 36–52.64), significantly higher than one month after recovery (27.37% (95% CI 19.97–37.53), ratio 0.62 (95% CI 0.44–0.87). By 6 months, avidity percentage increased in both vaccinated and recovered groups to 58.01% (95% Cl 51.26–65.64) and 43.53% (95% CI 34.01–55.70), respectively. Avidity further increased to 49.51 (95% Cl 43.69–56.09) in recovered individuals 12 months after diagnosis.

We compared the potentially more vulnerable populations to the less vulnerable (Figure 3A–F). We did not observe a difference in the immune response to vaccination between these groups; i.e., older and younger individuals, higher vs. lower BMI, or male vs. female, had similar peak antibody levels and a similar waning slope.

Yet, the immune response in the recovered individuals was different in the more vulnerable populations; older recovered patients had a significantly higher peak than younger recovered patients (2.45 (95% CI 2.15–2.76) vs. 1.71 (95% CI 1.46–1.97)), but both age groups had a similar waning slope (Figure 3A,B and Appendix A). Recovered patients with a higher BMI also had a significantly higher peak titer than recovered patients with a BMI <30 (2.94 (95% CI 2.49–3.37) vs. 1.81 (95% CI 1.57–2.04)). However, the slope of those with higher BMI was also slightly but significantly steeper (faster waning) (Figure 3C,D and Appendix A). Only slight, non-significant differences were noted between females and males, whether vaccinated or infected (Figure 3E,F and Appendix A). The results of log-transformed NAbs are similar to the results for the predicted values of log RBD-binding IgG (Figure 4), with a few differences. Older recovered patients had a significantly higher peak than younger recovered patients (5.31 (95% CI 4.96–5.66) vs. 4.87 (95% CI 4.56–5.18)), but both age groups had similar waning slopes (Figure 4A,B and Appendix A). Recovered patients with higher BMI also had a significantly higher peak titer than recovered patients with BMI <30 (5.55 (95% CI 5.01–6.10) vs. 4.93 (95% CI 4.64–5.23)), but both BMI groups had similar waning slopes (Figure 4C,D and Appendix A). Only slight, non-significant differences were noted between females and males, whether vaccinated or infected (Figure 4E,F and Appendix A). The models that compare the IgG and NAb kinetics of vaccinated and recovered individuals by different sub-populations (younger vs. older, BMI < 30 vs. BMI > 30, males/females) are presented in Appendix A. The plots of predicted values of log RBD-binding IgG are presented separately for each subgroup population in Figure 3A–F and for log transformed NAbs in Figure 4A–F.

Since we suspected that the reason for a higher response among the recovered patients with higher risk (older than 45 and those with BMI > 30) was due to a more severe infection in these groups, we performed a sub-analysis of recovered individuals by their severity of infection. The severity of infection was not related to differences in humoral response at the peak period in these groups (Appendix A).

Of the 120 recovered patients, the acute disease was asymptomatic in 9 individuals; 5 had severe disease; 16 had moderate disease; and most (*n* = 90) reported a mild acute disease. Among the moderately symptomatic individuals, one developed myocarditis.

Of the 120 recovered individuals, 116 responded to the follow-up questionnaire. In total, 42/116 (36.2%) individuals presented various manifestations of long-COVID (defined as persistent symptoms for >six weeks after recovery). Among them, respiratory symptoms were reported by 13/42 (30.9%), manifesting mostly as shortness of breath; 4/42 (9.5%) reported neurological manifestations, such as memory loss and concentration difficulties; 2/42 (4.8%) described mental symptoms as anxiety; and 25/42 (59.5%) complained of various fluctuating pain and discomfort manifestations. From the 42 individuals experiencing long-COVID symptoms, 37 (88%) were female. Although long-COVID symptoms were reported by a third of the population, we could not find an association with humoral kinetics, i.e., no differences in peak values of humoral response or in their slopes over time.

## 4. Discussion

In this study, we investigated the kinetics of the antibody titers following exposure to the vaccine or to infection among naïve individuals (i.e., those who had never been exposed to SARS-CoV-2 before). We assessed RBD-binding IgG and Nab titers on a monthly basis with a 6-month follow-up in a group of patients recovered from the ancestral SARS-CoV-2 and a matched group of individuals vaccinated with two doses of BNT162b2.

While several previous studies have also attempted to compare the immune response in such populations, they often involved small cohorts and examined differences in humoral immune dynamics between vaccinated and recovered individuals using combinations of unmatched cohorts (e.g., infected, re-infected, one-dose vaccine, two doses), which may complicate interpretation and further recommendation. [10,26,27,28,29,30].

We show that peak neutralizing and RBD-binding IgG antibodies after infection were lower than after vaccination, yet antibody waning was faster for the latter, thus suggesting a biological explanation for the longer protection from reinfection among recovered patients compared to the protection from breakthrough infections among vaccinated individuals [26,27,28,29,30].

In both the vaccinated and recovered groups, avidity increased with time, though it was relatively low even at the end of the study. Struck and colleagues [31] also reported relatively low avidity in recovered and vaccinated individuals. We have recently reported that a third vaccine dose results in significant and rapid increased avidity, which probably contributes to the superior immunogenicity and effectiveness compared to a second dose. This together may suggest that repeated exposure, either by three vaccine doses or hybrid immunity of infection and a vaccine dose, is needed to achieve higher avidity and eventually sustained and robust immunity [25,32,33]. While we only studied the humoral response, the differences between vaccinated and recovered individuals could be due to a different cellular response, potentially due to the role of the initial innate immune response as shown during the early stage of infection [34,35], or the mucosal response induced in infected but not in vaccinated individuals.

Interestingly, we observed higher peak RBD-binding IgG and neutralizing antibodies in recovered older individuals compared with younger individuals, regardless of their disease severity. Chronic inflammation is highly evident during aging, and therefore, a potential explanation of this phenomenon, which was also observed by Li and colleagues [36], is that older patients have greater activation of their immune system during recovery. Our results also demonstrate that another vulnerable population, individuals with BMI > 30, have higher RBD-binding IgG and neutralizing antibodies following infection compared with non-obese individuals. Since obesity is a state of chronic low level inflammation and people with obesity (PwO) have elevated levels of pro-inflammatory adipocyte-derived cytokines (adipokines), such as IL-1, IL-6, leptin, MCP-1, and TNF-α, and under-secretion of anti-inflammatory adipokines, such as adiponectin and IL-10 [37,38,39,40,41], it is possible that, similarly to older individuals, PwO have greater activation of their immune system during recovery. Future studies should investigate the differences in the immune response-specific pathways following vaccination and infection, which could be very important in elucidating factors that may be involved in antibody impairment in vulnerable populations, such as obese and older populations.

Another finding of our study is the rate of long-COVID reported, regardless of disease severity. As data accumulates, the extent and implications of long-term symptoms after SARS-COV-2 recovery are being revealed, presenting an additional economic and healthcare burden [42,43,44]. Most of the participants in our study suffered from mild or moderate disease symptoms (75% and 13%, respectively), yet more than a third continued to suffer from various symptoms for more than six weeks to several months after recovery. Most symptoms were respiratory, manifested mostly as shortness of breath or chronic coughing, as reported by others [44,45]. Interestingly, we found that female patients were most vulnerable to long-COVID symptoms, corresponding with Menges and colleagues [42]. Whether our relatively high long-COVID rates are strain-dependent and were observed due to infection with the ancestral strain or early VOCs is yet to be studied.

Our study has several limitations. First, the two groups were somewhat different, despite our attempt to match; the vaccinated cohort who were matched to the recovered cohort were HCW from the medical center, of which a majority were females, and thus could not be matched by sex. Furthermore, we did not match for comorbidities, yet the number of comorbidities per volunteer in both groups was similar. Only a minority of participants were immunosuppressed; however, the proportion of these was higher among the vaccinated group. Second, our recovered cohort included mostly mild cases with only five severe cases, restricting us from evaluating severity as a predictor for a sustainable immune response. Third, the follow-up duration of the two groups differed. While the infected group was followed for 1 year, vaccination was available just as of December 2020, and follow-up of the vaccinated group was extended for only 6 months. We therefore compared the two groups only during the first 6-month follow-up. Another limitation is that we studied only the humoral response, and potentially, the different response in infected vs. vaccinated people could be mostly due to the innate response, which induces different cellular responses. Last, these results cannot be generalized to the whole population since the study included mostly younger individuals and those infected with mild disease.

## 5. Conclusions

In conclusion, we report that peak antibody titers in vaccinated individuals were initially higher than recovered, yet waning was faster in the vaccinated. Thus, binding and neutralizing antibody titers persisted over time among recovered individuals. This may imply the benefit of hybrid immunity. While we observed higher antibody titers among recovered individuals at risk (older and obese), it is yet to be determined whether this has clinical significance for protection from reinfection. Further studies, assessing the differences in the immune response, both in the innate and cellular responses, between vaccinated and recovered patients are required.

## Figures and Tables

**Figure 1 microorganisms-11-01628-f001:**
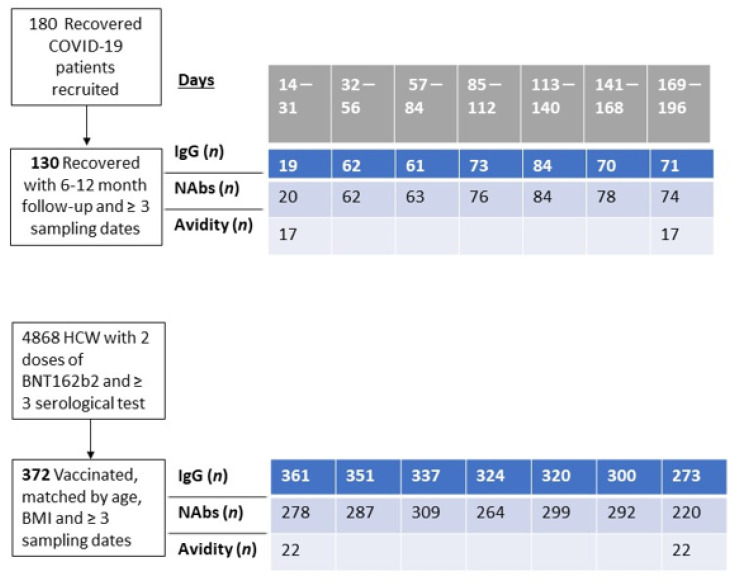
Study design and number of individuals followed.

**Figure 2 microorganisms-11-01628-f002:**
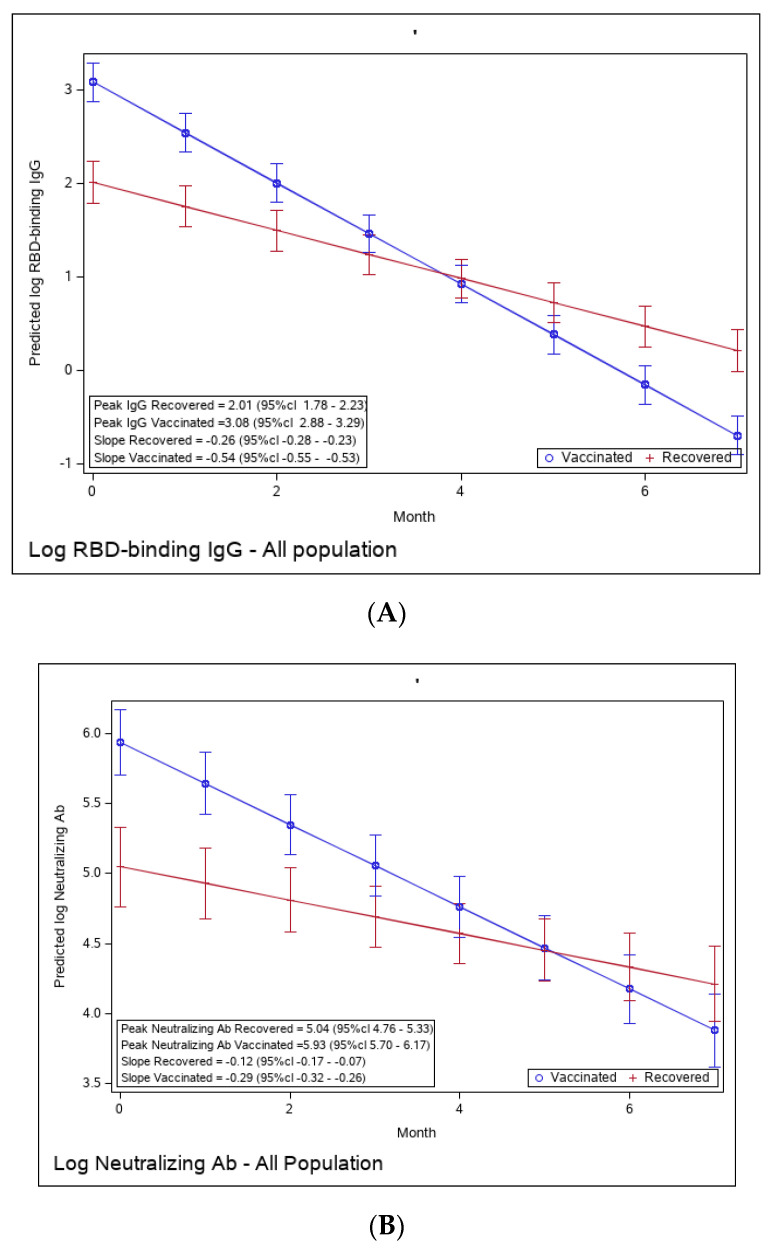
(**A**): Predicted log anti-RBD IgG of recovered versus vaccinated subjects. (**B**): Predicted log neutralizing Ab of recovered versus vaccinated subjects. (**C**): Avidity of RBD-binding IgG (%) of recovered versus vaccinated subjects. Predicted log anti-RBD IgG (**A**), predicted log neutralizing Ab (**B**), and % avidity of RBD-binding IgG (**C**) of recovered versus vaccinated subjects. The graph shows predicted values of log IgG for females with no comorbidities.

**Figure 3 microorganisms-11-01628-f003:**
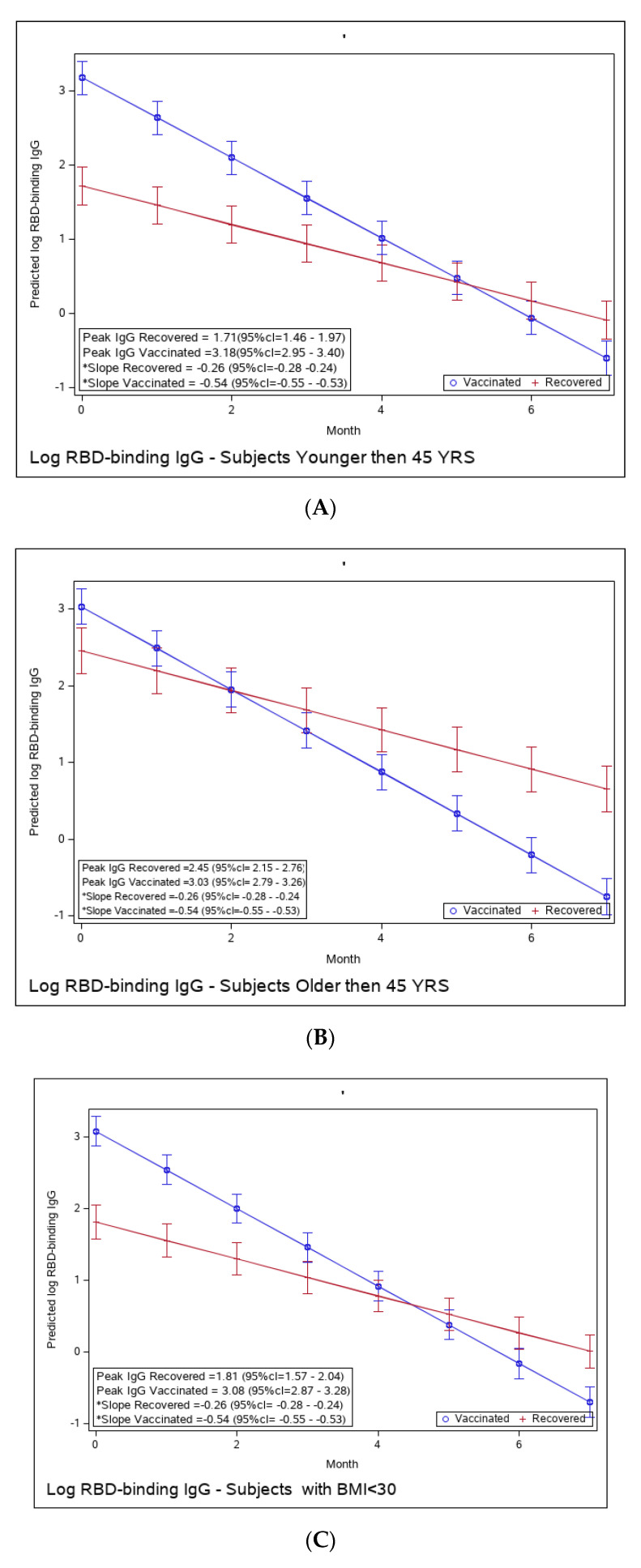
(**A**): Predicted log anti-RBD IgG of recovered versus vaccinated young adults (younger than 45). (**B**): Predicted log anti-RBD IgG of recovered versus vaccinated old adults (older than 45 years old). (**C**): Predicted log anti-RBD IgG of recovered versus vaccinated subjects with BMI < 30%. (**D**): Predicted log anti-RBD IgG of recovered versus vaccinated subjects with BMI > 30%. (**E**): Predicted log anti-RBD IgG of recovered versus vaccinated females. (**F**): Predicted log anti-RBD IgG of recovered versus vaccinated males. * The graphs show predicted values of log IgG for females with no comorbidities. The interaction between age group and period was insignificant, i.e., there is no difference in slope between young and old recovered subjects and between young and old vaccinated subjects. The interaction between BMI group and period was insignificant, i.e., there is no difference in slope between BMI < 30 and BMI > 30 recovered subjects and between BMI < 30 and BMI > 30 vaccinated subjects. The interaction between sex and period was insignificant, i.e., there was no difference in slope between females and males recovered subjects or between females and males vaccinated subjects.

**Figure 4 microorganisms-11-01628-f004:**
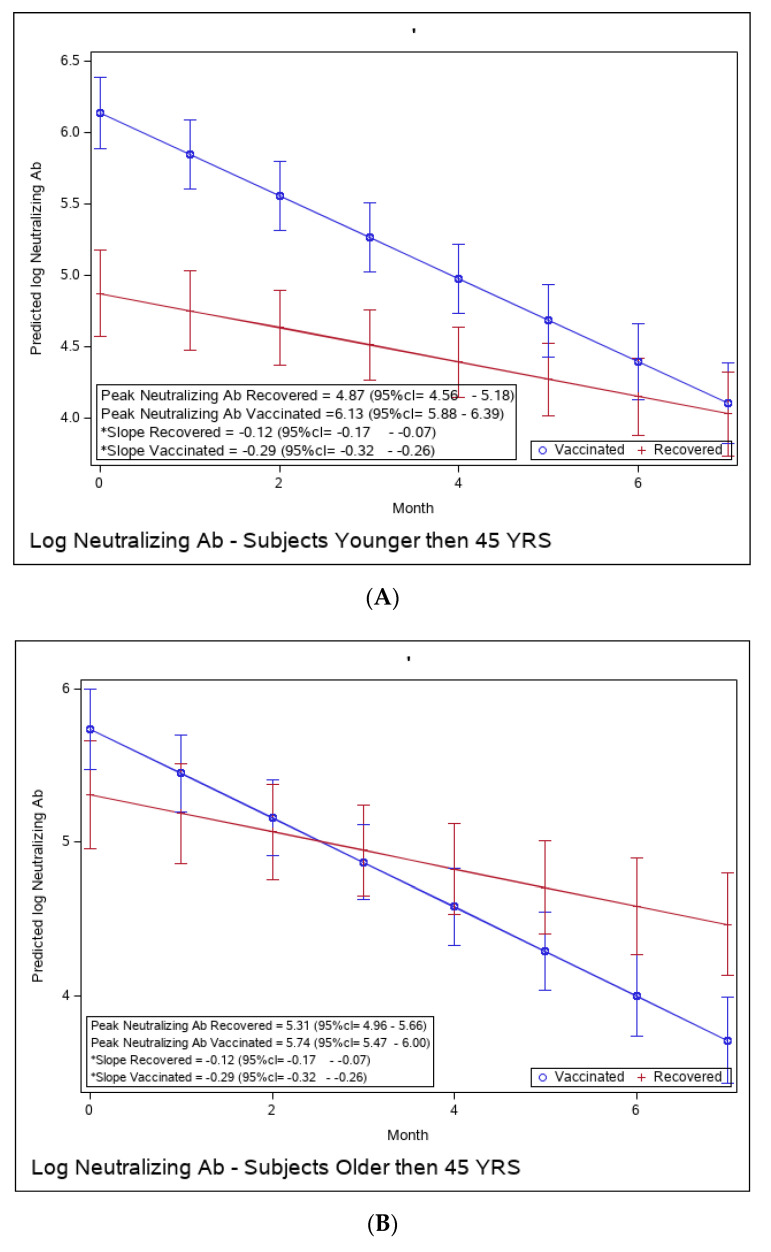
(**A**): Predicted log neutralizing Ab of recovered versus vaccinated young adults (younger than 45). (**B**): Predicted log neutralizing Ab of recovered versus vaccinated adults older than 45. (**C**): Predicted log neutralizing Ab of recovered versus vaccinated subjects with BMI < 30%. (**D**): Predicted log neutralizing Ab of recovered versus vaccinated subjects with BMI > 30%. (**E**): Predicted log neutralizing Ab of recovered versus vaccinated females. (**F**): Predicted log neutralizing Ab of recovered versus vaccinated males. * The graphs show predicted values of log IgG for females with no comorbidities. The interaction between age group and period was insignificant, i.e., there is no difference in slope between young and old recovered subjects and between young and old vaccinated subjects. The interaction between BMI group and period was insignificant, i.e., there is no difference in slope between BMI < 30 and BMI > 30 recovered subjects and between BMI < 30 and BMI > 30 vaccinated subjects. The interaction between sex and period was insignificant, i.e., there was no difference in slope between females and males recovered subjects or between females and males vaccinated subjects.

**Table 1 microorganisms-11-01628-t001:** Study population.

		Recovered	Vaccinated
Variable	N	120	372
	Number of controls matched per case mean (SD)		3.10 (1.21)
Age	Mean (SD)	39.70 (14.09)	41.69 (13.06)
	% > 45	41.66%	41.66%
BMI	% > 30	15.83%	15.83%
Sex	% Females	69 (57.50%)	306 (82.26%)
Comorbidities *	Mean of comorbidities (SD)	0.23 (+/−0.53)	0.38 (+/−0.71)
	0 N, (%)	98 (81.67%)	268 (72.04%)
	1 N, (%)	16 (13.33%)	72 (19.35%)
	>2 N, (%)	6 (5.00%)	32 (8.60%)
	Immunosuppressed	3 (2.5%)	40 (10.75%)
Disease Severity	Asymptomatic N, (%)	9 (7.5%)	-
	Mild symptoms N, (%)	90 (75%)	-
	Moderate symptoms N, (%)	16 (13.33%)	-
	Severe symptoms N, (%)	5 (4.17%)	-

* In the cohort of the vaccinated subjects, the comorbidities were: cardiac disease (*n* = 5), pulmonary illness (*n* = 13), immunosuppressed (*n* = 40), dyslipidemia (*n* = 23), diabetes (*n* = 19), and hypertension (*n* = 44). In the cohort of the recovered subjects, the comorbidities were: cardiac disease (*n* = 6), pulmonary illness (*n* = 2), immunosuppressed (*n* = 3), malignancy (*n* = 1), dyslipidemia (*n* = 3), diabetes (*n* = 6), and hypertension (*n* = 8).

## Data Availability

The data presented in this study are available on request from the corresponding author. The data are not publicly available due to restrictions eg privacy or ethical.

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
