# Peer review of "Humoral Immunity of Unvaccinated COVID-19 Recovered vs. Naïve BNT162b2 Vaccinated Individuals: A Prospective Longitudinal Study"

_microorganisms, 2023, doi:10.3390/microorganisms11071628_

Round 1

Reviewer 1 Report

The authors (AA) aim to characterize the humoral immune kinetics of SARS-CoV-2 recovered patients and vaccinated individuals.

This is an engaging article with a study design appropriate and useful to increase our knowledge of the issue. The title reports the key features of the paper encouraging the reader to read more.

Introduction

Lines 39-43: AA could add some references, such as:

Hall, V.J.; Foulkes, S.; Saei, A.; Andrews, N.; Oguti, B.; Charlett, A.; Wellington, E.; Stowe, J.; Gillson, N.; Atti, A.; et al. COVID-19 vaccine coverage in health-care workers in England and effectiveness of BNT162b2 mRNA vaccine against infection (SIREN): A prospective, multicentre, cohort study. Lancet 2021, 397, 1725–1735.

Modenese, A.; Paduano, S.; Bellucci, R.; Marchetti, S.; Bruno, F.; Grazioli, P.; Vivoli, R.; Gobba, F.; Bargellini, A. Investigation of Possible Factors Influencing the Neutralizing Anti-SARS-CoV-2 Antibody Titer after Six Months from the Second Vaccination Dose in a Sample of Italian Nursing Home Personnel. Antibodies 2022, 11, 59.

Modenese, A.; Paduano, S.; Bargellini, A.; Bellucci, R.; Marchetti, S.; Bruno, F.; Grazioli, P.; Vivoli, R.; Gobba, F. Neutralizing Anti-SARS-CoV-2 Antibody Titer and Reported Adverse Effects, in a Sample of Italian Nursing Home Personnel after Two Doses of the BNT162b2 Vaccine Administered Four Weeks Apart. Vaccines (Basel). 2021 Jun 15;9(6):652.

Favresse, J.; Bayart, J.L.; Mullier, F.; Dogné, J.M.; Closset, M.; Douxfils, J. Early antibody response in healthcare professionals after two doses of SARS-CoV-2 mRNA vaccine (BNT162b2). Clin. Microbiol. Infect. 2021 Sep;27(9):1351.e5-1351.e7.

Materials and methods

Lines 67, 81 and Figure 1: AA clarify the number of recovered patients. Are they 130 or 120 subjects?

Lines 163-167: AA should better explain the performed analysis.

Results

Table 1: what do the authors mean by the number of sets? AA should change gender with sex.

Figure 2, 3 and 4: Explain x-axis (Period). Are months? Moreover, what is the meaning of the following sentence “*The graph shows predicted values of Log IgG for females with no comorbidities.”? And why?

Lines 240 241: there are some differences in the trend between log Nabs and log RBD-binding IgG regarding the graphs stratified for age and sex. AA should better report these results.

Lines 301-304: AA should move this sentence from the results to the discussion section.

Lines 306-316: AA should better report the data. Are 42 patients with moderate symptoms or are 42 subjects with various manifestation of long- COVID? Or both?

Discussion

AA should comment on the association with comorbidities

Conclusion

Lines 383-388: the authors should show more caution in their conclusions regarding vaccination and its protection from the risk of getting infected and getting sick, also given the vaccinal hesitation that exists at the moment and above all in light of the limitations of their study.

Supplementary materials

Delete comments

Author Response

Reviewer # 1

We thank you for your prompt and constructive review. Below are the answers relating to all the comments:

Comments and Suggestion for Authors

The authors (AA) aim to characterize the humoral immune kinetics of SARS-CoV-2 recovered patients and vaccinated individuals.

This is an engaging article with a study design appropriate and useful to increase our knowledge of the issue. The title reports the key features of the paper encouraging the reader to read more.

Introduction

  1. Lines 39-43: AA could add some references, such as:

Hall, V.J.; Foulkes, S.; Saei, A.; Andrews, N.; Oguti, B.; Charlett, A.; Wellington, E.; Stowe, J.; Gillson, N.; Atti, A.; et al. COVID-19 vaccine coverage in health-care workers in England and effectiveness of BNT162b2 mRNA vaccine against infection (SIREN): A prospective, multicentre, cohort study. Lancet 2021, 397, 1725–1735.

Modenese, A.; Paduano, S.; Bellucci, R.; Marchetti, S.; Bruno, F.; Grazioli, P.; Vivoli, R.; Gobba, F.; Bargellini, A. Investigation of Possible Factors Influencing the Neutralizing Anti-SARS-CoV-2 Antibody Titer after Six Months from the Second Vaccination Dose in a Sample of Italian Nursing Home Personnel. Antibodies 2022, 11, 59.

Modenese, A.; Paduano, S.; Bargellini, A.; Bellucci, R.; Marchetti, S.; Bruno, F.; Grazioli, P.; Vivoli, R.; Gobba, F. Neutralizing Anti-SARS-CoV-2 Antibody Titer and Reported Adverse Effects, in a Sample of Italian Nursing Home Personnel after Two Doses of the BNT162b2 Vaccine Administered Four Weeks Apart. Vaccines (Basel). 2021 Jun 15;9(6):652.

Favresse, J.; Bayart, J.L.; Mullier, F.; Dogné, J.M.; Closset, M.; Douxfils, J. Early antibody response in healthcare professionals after two doses of SARS-CoV-2 mRNA vaccine (BNT162b2). Clin. Microbiol. Infect. 2021 Sep;27(9):1351.e5-1351.e7.

Answer: the suggested references were added to the introduction.

Materials and methods

  1. Lines 67, 81 and Figure 1: AA clarify the number of recovered patients. Are they 130 or 120 subjects?

Answer: 130 recovered patients were recruited and analyzed in the avidity test. Only for 120 of the 130 recovered individuals, we could find matched vaccinated individuals. A sentence was added at the end of the Study design and population chapter:" From the 130 recovered individuals, only 120 could be matched to BNT162b2 vaccinated individuals.

  1. Lines 163-167: AA should better explain the performed analysis.

Answer- An explanation of the analysis was added to the paragraph, this is the changed paragraph: To evaluate the functionality of antibodies, we tested the RBD-binding IgG avidity score at one- and six-months post vaccination and one, six- and 12-months following recovery. Avidity was calculated as the ratio (in percentage) between sample OD with 6 M urea and sample OD with PBS.  One month after vaccination, avidity GMT was 43.53% (95%cl 36 - 52.64) significantly higher than one month after recovery (27.37% (95%CI 19.97 - 37.53), ratio 0.62 95%CI 0.44-0.87). By six-months, avidity percentage increased in both vaccinated and recovered groups with   58.01% (95%cl 51.26-65.64), and 43.53% (95% CI 34.01-55.70), respectively. Avidity further increased to 49.51 (95%cl 43.69-56.09) in recovered individuals 12 months after diagnosis.

Results

  1. Table 1: what do the authors mean by the number of sets? AA should change gender with sex.

Answer- A set is a recovered individual matched with 1-6 vaccinated individuals serving as controls. But you are right it is confusing, thus we deleted this row from table 1. 

The word Gender in table 1 was changed to sex.

  1. Figure 2, 3 and 4: Explain x-axis (Period). Are months? Moreover, what is the meaning of the following sentence “*The graph shows predicted values of Log IgG for females with no comorbidities.”? And why?

Answer – All title of all x-axis were changed to month.  The x- axis represents the periods blood was taken. Blood was taken every month from each individual, each person with its' own schedule/timetable. The periods represent the range of days' blood was taken so from each individual the space between each blood taking was one month.

The sentence: “*The graph shows predicted values of Log IgG for females with no comorbidities,” explains the model we used. Graphs of population-based predicted values for vaccinated versus recovered subjects were generated from these models, with females with no comorbidities as the reference.

  1. Lines 240 241: there are some differences in the trend between log Nabs and log RBD-binding IgG regarding the graphs stratified for age and sex. AA should better report these results.

Answer- We agree and have now addressed this in a paragraph regarding the results of Nab that was added to the results:" The results of log transformed NAbs are similar to the results for the predicted values of log RBD-binding IgG (Figure 4) with a few differences. Older recovered patients had a significantly higher peak than younger recovered (5.31 (95%CI 4.96-5.66) vs. 4.87 (95%CI 4.56-5.18)), but both age groups had a similar waning slope (Fig 4A,4B and Table S9). Recovered patients with higher BMI also had a significantly higher peak titer than recovered patients with BMI <30 (5.55 (95%CI 5.01-6.10) vs. 4.93 (95%CI 4.64 -5.23)), but both BMI groups had a similar waning slope (Fig 4C, 4D and Table S10). Only slight non-significant differences were noted between females and male, whether vaccinated or infected (Fig 4E, 4F, and Table S11)."

  1. Lines 301-304: AA should move this sentence from the results to the discussion section

Answer- In this sentence we report the results of a sub analysis assessing the interaction between severity and age group and interaction between severity and BMI group. A Linear mixed model of log RBD - binding IgG and log NAbs including interaction between severity and Age group was used as well as a linear mixed model of log RBD - binding IgG and log NAbs including interaction between severity and BMI group. The detailed results are presented in Table S12 and S13.

  1. Lines 306-316: AA should better report the data. Are 42 patients with moderate symptoms or are 42 subjects with various manifestation of long- COVID? Or both?

Answer- Among the recovered individuals 42 reported to have long-COVID symptoms. The last section in the results was slightly confusing, we have now clarified, and rephrased it:

"Of the 120 recovered patients, the acute disease was asymptomatic in nine individuals, five had severe disease, 16 had moderate disease, and most (n=90) reported a mild acute disease. Among the moderately symptomatic individuals one developed myocarditis.

Of 120 recovered individuals, 116 responded to the follow-up questionnaire. In total, 42/116 (36.2%) individuals presented various manifestation of long-COVID (defined as persistent symptoms for >six weeks after recovery). Among them, respiratory symptoms were reported by 13/42 (30.9%) manifested mostly as shortness of breath. 4/42 (9.5%) reported neurological manifestation as memory loss and concentration difficulties. 2/42 (4.8%) described mental symptoms as anxiety and 25/42 (59.5%) complained of various fluctuating pain and discomfort manifestation. From the 42 individuals experiencing long-COVID symptoms, 37 (88%) were females. Although Long-COVID symptoms were reported by a third of the population, we could not find association to humoral kinetics, i.e no differences in peak values of humoral response and in their slopes over time."

Discussion

  1. AA should comment on the association with comorbidities

Answer- We assessed the association with comorbidities, but did not find any significant result and therefore did not present this. We feel it is inappropriate to discuss this, since it might just be due to lack of power.

Conclusion

  1. Lines 383-388: the authors should show more caution in their conclusions regarding vaccination and its protection from the risk of getting infected and getting sick, also given the vaccinal hesitation that exists at the moment and above all in light of the limitations of their study.

Answer- We have changed the conclusions to make it more cautious:" In conclusion, our study addresses several important aspects comparing exposure to SARS-CoV-2 virus or vaccine. The peak antibody titers of vaccinated individuals were higher than recovered, yet waning was faster in the vaccinated. Thus, the persistent levels of antibodies among recovered individuals might protect them further from being re-infected over time. This may imply the benefit of hybrid immunity.

Supplementary materials

  1. Delete comments

Answer- Thank you, comments were deleted.

Reviewer 2 Report

Comments and suggestions

Title: Humoral immunity of unvaccinated COVID-19 recovered vs. naïve BNT162b2 vaccinated individuals: A prospective longitudinal study

Article ID: microorganisms-2413870

Summary section:

1. The conclusions in this section are not in accordance with what is suggested in the title or the objectives

2. Throughout the summary, "immune response" is mentioned, this is very general since the present study only evaluates one of the components of this system, such as neutralizing antibodies. Put this in the abstract.

3. Homogenize the correct name COVID-19 and not just COVID.

4. In keywords review and place only MeSH terminology

Introduction Section

5. The first paragraph is repetitive and very general, not to say outdated. In addition, it is not an adequate introduction to the topic raised.

6. Expand the introduction by focusing on neutralizing antibodies.

methods section

7. Was the diagnosis of COVID-19 by RT-PCR in real time? Was the same diagnostic kit used for all enrolled patients?

8. Because the group of vaccinated health workers was chosen compared to the group of recovered from which it is not mentioned if they were health personnel or the general population. This is important because populations are not homogeneous, health personnel are more exposed to the disease than the general population.

9. Justify why the periodicity of the analyzes was monthly.

10. Because the 12-month follow-up was not completed in the vaccinated group

11. Did they evaluate any other comorbidity?

Results section:

12. The authors state that they found long-lasting rates of COVID-19. If no PCR control or viral viability study was performed such as persistent COVID-19 can be excluded. In any case, the definition and the way to determine long-term COVID-19 should be considered in the methodology section.

Conclusion section:

13. Place the protection times according to the results of each group.

Minor editing of English language required 

Author Response

Reviewer #2

We thank you for your prompt and constructive review. Below are the answers relating to all the comments:

Comments and suggestions

Title: Humoral immunity of unvaccinated COVID-19 recovered vs. naïve BNT162b2 vaccinated individuals: A prospective longitudinal study

Article ID: microorganisms-2413870

Summary section:

1.The conclusions in this section are not in accordance with what is suggested in the title or the objectives

Answer: We agree that the summary was not well phrased to represent the results and conclusion, we have now rephrased it:

To better understand the differences in the immune response to SARS-CoV-2 infection compared to the response to vaccination, we characterized the humoral immune kinetics of these populations.

In this prospective longitudinal study, we followed unvaccinated COVID-19 recovered individuals (n=130) and naïve, two dose BNT162b2 vaccinated individuals (n=372) who were age- and BMI-matched, for six months during the first COVID-19 pandemic year.

Anti-RBD-IgG, neutralizing antibodies (NAbs) and avidity were compared monthly for six months. For recovered patients, data on symptoms and severity of the disease were collected.

Anti-RBD-IgG and NAbs titers at peak were higher after vaccination vs. after infection, but the decline was steeper (peak log IgG: 3.08 vs. 1.81, peak log NAbs:5.93 vs. 5.04, slopes: -0.54 vs. -0.26). Peak anti-RBD-IgG and NAbs were higher in recovered individuals with BMI>30 and in older individuals compared to individuals with BMI<30, younger population. Of the recovered, 42 (36%) experienced long-COVID symptoms. Avidity was initially higher in vaccinated compared with recovered individuals, though with time, it increased in recovered but not among vaccinated.

Here, we show that while the initial antibody titers, neutralization and avidity are lower in SARS-CoV-2-recovered individuals, they persist for longer duration. These results suggest a differential protection against COVID-19 in recovered unvaccinated vs. naive vaccinated individuals.

  1. Throughout the summary, "immune response" is mentioned, this is very general since the present study only evaluates one of the components of this system, such as neutralizing antibodies. Put this in the abstract.

Answer: We have added both in the description of the Methods, the Results and the Conclusion, that we address antibody titers, neutralization and avidity.

  1. Homogenize the correct name COVID-19 and not just COVID.

Answer- All mention of COVID, is now COVID-19, and at some referrals, where we refer to the virus, we use "SARS-CoV-2". 

  1. In keywords review and place only MeSH terminology

Answer: The keywords were changed according to MeSH terminology:

Keywords: Immunoglobulin G; Antibodies, Neutralizing; Immunity, Humoral; COVID-19; Infection; BNT162b2 vaccine

Introduction Section

  1. The first paragraph is repetitive and very general, not to say outdated. In addition, it is not an adequate introduction to the topic raised.

Answer: We have rephrased the introduction to address this comment.

  1. Expand the introduction by focusing on neutralizing antibodies.

Answer: We have now further emphasized the importance of the neutralizing antibodies in the introduction.

methods section

  1. Was the diagnosis of COVID-19 by RT-PCR in real time? Was the same diagnostic kit used for all enrolled patients?

Answer: The diagnosis of COVID-19 by RT-PCR was in real time, but at the time, several diagnostic kits were used ; Allplex™ 2019-nCoV (Seegene, S. Korea), NeuMoDx™ SARS-CoV-2 assay (NeuMoDx™ Molecular, AnnArbor, Michigan), Xpert®, Xpress SARS-CoV-2 (Cepheid, Sunnyvale, CA, USA). All tests were performed according to manufacturers' instructions

  1. Because the group of vaccinated health workers was chosen compared to the group of recovered from which it is not mentioned if they were health personnel or the general population. This is important because populations are not homogeneous, health personnel are more exposed to the disease than the general population.

Answer- Indeed the two groups are slightly different. The vaccinated group includes only HCW, while the recovered were from the general population, with only a small fraction of HCW. To overcome this, we adjusted for comorbidities, sex and age. We do not believe that exposure among the groups was different, according to our previous studies, showing that the majority of HCW infections were due to community exposure and not hospital exposure (probably due to appropriate use of PEP).

  1. Justify why the periodicity of the analyzes was monthly.

Answer: Blood was taken every month from each individual, each person with its' own schedule/timetable. The periods represent the range of days' blood was taken so from each individual the space between each blood taking was one month. All title of x-axis was changed to month, in order to clarify. 

  1. Because the 12-month follow-up was not completed in the vaccinated group

Answer: For the vaccinated group, we had only a 6 months' follow-up, since most of the HCW then received a booster, third dose.

  1. Did they evaluate any other comorbidity?

Answer: All of the following comorbidities and medical conditions were assessed: Hypertension, diabetes, dyslipidemia, heart disease, lung disease, kidney disease, coagulation disorders, liver disease, pregnancy, allergies, autoimmune disease, malignancies, immunosuppression including: biological therapy, chemotherapy, splenectomy any organ transplantation and HIV. The comorbidities were gathered into groups, which have been shown to be associated with greater risk for COVID-19 infection, and only these were included in the analyses. The number of participants with any of these comorbidities is described in the legend of Table 1: In the cohort of the vaccinated subjects: Cardiac disease(n=5), Pulmonary illness (n=13), Immunosuppressed (n=40), Dyslipidemia (n=23), Diabetes (n=19), Hypertension (n=44). In the cohort of the recovered subjects: Cardiac disease(n=6), Pulmonary illness (n=2), Immunosuppressed (n=3), Malignancy (n=1) Dyslipidemia (n=3), Diabetes (n=6), Hypertension (n=8).

Results section:

  1. The authors state that they found long-lasting rates of COVID-19. If no PCR control or viral viability study was performed such as persistent COVID-19 can be excluded. In any case, the definition and the way to determine long-term COVID-19 should be considered in the methodology section.

Answer: The last section in the results was slightly confusing, we have now clarified, and rephrased it:

"Of the 120 recovered patients, the acute disease was asymptomatic in nine individuals, five had severe disease, 16 had moderate disease, and most (n=90) reported a mild acute disease. Among the moderately symptomatic individuals one developed myocarditis.

Of 120 recovered individuals, 116 responded to the follow-up questionnaire. In total, 42/116 (36.2%) individuals presented various manifestation of long-COVID (defined as persistent symptoms for >six weeks after recovery). Among them, respiratory symptoms were reported by 13/42 (30.9%) manifested mostly as shortness of breath. 4/42 (9.5%) reported neurological manifestation as memory loss and concentration difficulties. 2/42 (4.8%) described mental symptoms as anxiety and 25/42 (59.5%) complained of various fluctuating pain and discomfort manifestation. From the 42 individuals experiencing long-COVID symptoms, 37 (88%) were females. Although Long-COVID symptoms were reported by a third of the population, we could not find association to humoral kinetics, i.e no differences in peak values of humoral response and in their slopes over time."

Conclusion section:

  1. Place the protection times according to the results of each group.

Answer: Since this study did not assess correlates of protection, we cannot define protection times. We show clearly that antibody titers persist with time among recovered and show slower waning, but stating protection times would be a bit pretentious. More so, with the evolving variants, we and others have shown, that immune thresholds change and protection times from the ancestral strain is probably very different from that of current VOCs. Nevertheless, we have rephrased the conclusion to better summarize the study:" In conclusion, we report that peak antibody titers of vaccinated individuals were initially higher than recovered, yet waning was faster in the vaccinated. Thus, binding and neutralizing antibody titers persisted over-time among recovered individuals.  This may imply the benefit of hybrid immunity. While we observed higher antibody titers among recovered individuals at risk (older and obese), it is yet to be determined whether this has clinical significance in protection from reinfection. Further studies, assessing the differences in the immune response both in the innate and cellular responses between vaccinated and recovered are required.

Minor editing of English language required 

Answer: Manuscript went through English language editing.

Round 2

Reviewer 1 Report

The authors (AA) have carefully addressed the reviewers' comments. Overall the changes made have improved the manuscript, but there few points to improve.

Discussion:

In the paragraph about limitations, AA should better explain the differences between the groups studied. They cannot generalize their results due to their selection criteria. Moreover, AA should underline the limit related to different follow-ups between vaccinated and recovered subjects.

Author Response

Dear reviewer,

We thank very much the reviewer for their prompt, efficient and constructive review and remarks to our manuscript.

We thank you for the opportunity to revise our manuscript. Here, we address each and every point raised by the reviewer:

Reviewer # 1

Discussion: 

In the paragraph about limitations, AA should better explain the differences between the groups studied. They cannot generalize their results due to their selection criteria. Moreover, AA should underline the limit related to different follow-ups between vaccinated and recovered subjects.

Answer: Thank you for this important remark. We have changed the limitation paragraph to address the differences between the groups studied, the generalization and the limit related to different follow-ups between vaccinated and recovered subjects. This is the revised paragraph:" Our study has several limitations. First, the two group were somewhat different, despite our attempt to match; the vaccinated cohort who were matched to the recovered cohort were HCW from the medical center, in which a majority were females, and thus could not be matched by sex. Furthermore, we did not match for comorbidities, yet, the number of comorbidities per volunteer, in both groups was similar. Only a minority of participants were immunosuppressed, however the proportion of these was higher among the vaccinated group. Second, our recovered cohort included mostly mild cases with only five severe cases, restricting us from evaluating severity as a predictor for sustainable immune response. Third, follow up duration of the two groups differed. While the infected group was followed for 1 year, vaccination was available just from Dec, 2020 and follow-up of the vaccinated group extended for only 6 months. We therefore compared the two groups only during the first 6-months follow-up Another limitation is that we studied only the humoral response, and potentially, the different response in infected vs. vaccinated could be mostly due to the innate response, which induces different cellular response. Last, these results cannot be generalized to the whole population, since the study included mostly younger individuals, and those infected, were mostly with mild disease."

Reviewer 2 Report

The authors have partially responded to comments and suggestions. However, in my opinion the manuscript still has important limitations. The differences between the groups studied eg. For this reason I do not consider it suitable for publication.

Minor editing of English language required

Author Response

Dear reviewer,

We thank very much the reviewer for the prompt, efficient and constructive review and remarks to our manuscript.

We thank you for the opportunity to revise our manuscript. Here, we address each and every point raised by the reviewer:

Reviewer 2

The authors have partially responded to comments and suggestions. However, in my opinion the manuscript still has important limitations. The differences between the groups studied eg. For this reason, I do not consider it suitable for publication.

 Answer: Thank you for this important remark. We have changed the limitation paragraph to address the differences between the groups studied, the generalization and the limit related to different follow-ups between vaccinated and recovered subjects. This is the revised paragraph:" Our study has several limitations. First, the two group were somewhat different, despite our attempt to match; the vaccinated cohort who were matched to the recovered cohort were HCW from the medical center, in which a majority were females, and thus could not be matched by sex. Furthermore, we did not match for comorbidities, yet the number of comorbidities per volunteer, in both groups was similar. Only a minority of participants were immunosuppressed, however the proportion of these was higher among the vaccinated group. Second, our recovered cohort included mostly mild cases with only five severe cases, restricting us from evaluating severity as a predictor for sustainable immune response. Third, follow up duration of the two groups differed. While the infected group was followed for 1 year, vaccination was available just from Dec, 2020 and follow-up of the vaccinated group extended for only 6 months. We therefore compared the two groups only during the first 6-months follow-up Another limitation is that we studied only the humoral response, and potentially, the different response in infected vs. vaccinated could be mostly due to the innate response, which induces different cellular response. Last, these results cannot be generalized to the whole population, since the study included mostly younger individuals, and those infected, were mostly with mild disease."